# Dipicolinate Complexes of Oxovanadium(IV) and Dioxovanadium(V) with 2-Phenylpyridine and 4,4′-Dimethoxy-2,2′-bipyridyl as New Precatalysts for Olefin Oligomerization

**DOI:** 10.3390/ma15041379

**Published:** 2022-02-13

**Authors:** Joanna Drzeżdżon, Marta Pawlak, Barbara Gawdzik, Aleksandra Wypych, Karol Kramkowski, Paweł Kowalczyk, Dagmara Jacewicz

**Affiliations:** 1Department of Environmental Technology, Faculty of Chemistry, University of Gdansk, Wita Stwosza 63, 80-308 Gdansk, Poland; marta.pawlak0812@gmail.com (M.P.); dagmara.jacewicz@ug.edu.pl (D.J.); 2Institute of Chemistry, Jan Kochanowski University, Uniwersytecka 7, 25406 Kielce, Poland; b.gawdzik@ujk.edu.pl; 3Centre for Modern Interdisciplinary Technologies Nicolaus Copernicus, University in Torun ul. Wileńska 4, 87-100 Toruń, Poland; wypych@umk.pl; 4Department of Physical Chemistry, Medical University of Bialystok, Kilińskiego 1, 15-089 Białystok, Poland; kkramk@wp.pl; 5Department of Animal Nutrition, The Kielanowski Institute of Animal Physiology and Nutrition, Polish Academy of Sciences, Instytucka 3, 05-110 Jabłonna, Poland

**Keywords:** oxovanadium(IV) complexes, dioxovanadium(V) complexes, olefin oligomerization, allyl alcohol, norbornene

## Abstract

Polyolefins are used in everyday life, including in the production of many types of plastic. In addition, polyolefins account for over 50% of the polymers produced in the world. After conducting the oligomerization reactions of 2-propen-1-ol, 2-chloro-2-propen-1-ol, and norborene, polyolefins are obtained. In this report, two complexes of oxovanadium(IV) and dioxovanadium(V) with dipicolinate, 2-phenylyridine, and 4,4′-dimethoxy-2,2′-bipyridyl as precatalysts for 2-propen-1-ol, 2-chloro-2-propen-1-ol, and norborene oligomerizations are prepared. We present for the first time the new dipicolinate complex compound of oxovanadium(IV) with 4,4′-dimetoxy-2,2′-bipyridyl. Both complexes were tested for catalytic activity in the oligomerization reactions of 2-propen-1-ol, 2-chloro-2-propen-1-ol, and norbornene. Both synthesized complexes showed high catalytic activity in these oligomerization reactions, except for the oligomerization of norbornene.

## 1. Introduction

In the past 30 years, the use of organometallic compounds of the d-block elements of the periodic table, such as vanadium [1], titanium [1], rhodium [2], tungsten [3], molybdenum [3] and cobalt [1,4], has resulted in the development of new, innovative, and selective methods for the synthesis of organometallic compounds.

Particularly noteworthy are Grubbs catalysts, which contain ruthenium ions as the central atoms in complex compounds and are characterized by their high catalytic activity in olefin metathesis reactions [5,6,7,8]. We refer to these types of catalysts as the “well-defined catalysts”, i.e., those that, in terms of the degree of oxidation of the metal ion and the coordination sphere of ligands, resemble the actual catalyst of the process, i.e., an individual participating in the catalytic cycle [9,10,11]. This type of catalyst is sufficiently stable, which enables its spectroscopic and x-ray structure characterization, which in turn makes it possible to observe and control the formation of intermediate products. Systematic studies of the effect of complex geometry, electronic and steric properties of ligands on reactivity, selectivity, thermal stability and functional group tolerance can be carried out for such complexes [12,13,14,15]. The high tolerance of Grubbs catalysts in relation to atmospheric oxygen and moisture, combined with a wide range of possible applications, e.g., cyclization processes, or oligomerization and polymerization processes, has made the metathesis process an important synthetic tool and one of the most important methods of obtaining carbon–carbon bonds in organic chemistry. The great importance of this type of reaction is evidenced by the award of the Nobel Prize in 2005 to Prof. Y. Chauvin, R. Schrock, and R.H. Grubbs “for the development of metathetical methods in organic synthesis”. The synthesis of ruthenium alkylidene complexes and the discovery of their catalytic activity in the processes of metathetic olefin conversion resulted in a large increase in the interest of scientists in this reaction and, consequently, caused an sudden increase in the number of applications of metathesis in organic synthesis [16,17,18,19,20,21,22,23,24]. From the point of view of practical application, especially on an industrial scale, it is highly desirable that the newly synthesized organometallic complexes of transition metal ions are characterized by high stability under elevated temperature conditions and that they can be purified and stored without a protective gas atmosphere. In some applications it is also important thatdepending on the reaction conditions, these catalysts show a delayed initiation and, after initiation, promote the reaction quickly enough [25,26,27,28,29,30].

In addition to the metal ion, the coordination sphere, i.e., ligands directly related to the central ion [31,32,33], has a very significant impact on the catalytic properties. The functional groups present in the substrate, as well as oxygen or water present in the reaction system, can influence the activity of the catalysts. These factors can react with the active center of the metal, resulting in a partial or complete loss of catalytic properties. In conclusion, further research is necessary to discover more active catalysts and with more universal application in relation to already well-known and defined catalysts.

This report is a continuation of our previous work on olefin oligomerization catalysis. In the present study, we prepared dipicolinate complexes of oxovanadium(IV) and dioxovanadium(V) with 2-phenylpyridine and 4,4′-dimethoxy-2,2′-bipyridyl. The novelty of this report is the presentation of a new, previously undescribed dipicolinate complex compound of oxovanadium(IV) with 4,4′-dimethoxy-2,2′-bipyridyl, and in this manuscript for the first time we describe the use of two complex compounds: [VOO (dipic)](2-phepyH) · H_2_O and [VO(dipic)(dmbipy)] · 2 H_2_O for the oligomerization of 2-chloro-2-propen-1-ol, 2-propen-1-ol, and norbornene. To the best of our knowledge, there is no information in the literature about the catalytic properties of the complexes: [VOO (dipic)](2-phepyH) · H_2_O and [VO(dipic)(dmbipy)] · 2 H_2_O.

## 2. Materials and Methods

### 2.1. Materials

All chemical compounds: vanadyl acetylacetonate, 2-phenylpyridine, 4,4′-dimethoxy-2,2′-bipyridine, toluene, dimethylsulfoxide, dipicolinic acid, modified methylaluminoxane (7% aluminum in toluene), 2-propen-1-ol, 2-chloro-2-propen-1-ol, and norbornene were purchased from Merck, Darmstad, Germany. The purity of reagents was in the range 97–100%.

### 2.2. Complex Compounds Synthesis

[VOO(dipic)](2-phepyH) H_2_O: In order to synthesize the compound, 2.65 g (0.01 mol) of vanadyl acetylacetonate VO(acac)_2_, 1.67 g (0.01 mol) of dipicolinic acid (H_2_dipic) and 2.86 (0.02 mol) of 2-phenylpyridine were mixed and to this mixture 50 mL of water has been added. The whole solution was heated to boiling in a heating mantle, after which the color of the solution changed from green-blue to green-yellow. After cooling, the solution was allowed to precipitate brown crystals of [VOO(dipic)](2-phepyH) · H_2_O.

[VO(dipic)(dmbipy)] 2 H_2_O: Vanadyl acetylacetonate VO(acac)_2_ (2.13 mmol, 0.57 g) was mixed with dipicolinic acid (2.15 mmol, 0.36 g) (H_2_dipic) and 4,4′-dimethoxy-2,2′-bipyridine (2.13 mmol, 0.46 g). In the next step, all reagents were dissolved in water (50 mL). The solution was heated under reflux for 2 h. After cooling the mixture, a light-green precipitate appeared, which was the dipicolinate oxovanadium(IV) complex compound with 4,4′-dimethoxy-2,2′-bipyridine.

### 2.3. Elemental Analysis

Elemental analysis (the Vario El Cube apparatus Langenselbold, Germany) was conducted on samples (2 mg) of the complexes that were homogeneous and dry.

### 2.4. IR Spectra

The IR spectra were recorded in the range 4000 cm^−1^–600 cm^−1^ on a KBr pastil. DLATGS (Branch Überlingen, Germany)was a detector. The IFS66 apparatus by BRUKER (Branch Überlingen, Germany ) has a resolution of 0.12 cm^−1^.

### 2.5. MALDI-TOF-MS Spectra

The MALDI-TOF-MS spectra were recorded on the Bruker Biflex III company, (Branch Überlingen, Germany). 2,5-Dihydroxybenzoic acid (DHB) and α-cyano-4-hydroxycinnamic acid (CCA) were used as a matrix.

### 2.6. Oligomerization Process

First, 3 µmol of the synthesized complex, which served as a precatalyst, was prepared and placed in a glass cell with a tightly closed stopper. The weighed complex was then dissolved in 1 mL of toluene and 1 mL of DMSO. The cell was placed in a stand, to which a nitrogen balloon and a syringe for air evacuation were connected, allowing the polymerization reaction to be carried out in a nitrogen atmosphere. In the next step, 3 mL of MMAO-12 solution was added dropwise into the cell, followed by 3 mL of 2-propen-1-ol, 2-chloro-2-propen-1-ol, or norbornene. The mixture was stirred over a magnetic stirrer until a gel was formed and it was constantly heated in a water bath at a temperature of 65 °C. When a gel was formed, the solution was washed with a mixture of hydrochloric acid and methanol in a 1:1 molar ratio.

## 3. Results and Discussion

New oxovanadium(IV) complex compound with dipicolinate and 4,4′-dimethoksy-2,2′-bipyridine (dmbipy) and dioxovanadium(V) dipicolinate complex with 2-phenylpyridine (2-phepy) with the formulas [VO(dipic)(dmbipy)] · 2 H_2_O and [VOO(dipic)](2-phepyH) · H_2_O were synthesized (Figure 1).

The compound [VOO(dipic)](2-phepyH) · H_2_O was obtained as crystals, while compound [VO(dipic)(dmbipy)] · 2 H_2_O was obtained as a powder. All crystal parameters of the complex have been published in the literature [34]. In [VOO(dipic)](2-phepyH) · H_2_O crystal structure, the lengths and angles of bonds characterizing the geometry of the dioxo-(pyridine-2,6-dicarboxylate)-vanadium(V) cation have values similar to other crystal structures containing this oxovanadium cation(V) [34]. XRD studies of the crystals of the synthesized compound, [VOO(dipic)](2-phepyH) · H_2_O, confirmed that this complex compound crystallizes in the monoclinic P21/c space group. An asymmetric unit consists of one 2-phenylpyridine cation, one dioxo (pyridine-2,6-dicarboxylate)anion)-vanadium(V), and one water molecule. The crystal structure [VOO(dipic)](2-phepyH) · H_2_O contains ions and water molecules bound by hydrogen bonds [34].

Elemental analysis confirmed the composition of the obtained [VO(dipic)(dmbipy)] · 2 H_2_O. The results of elemental analysis of [VO(dipic)(dmbipy)] · 2 H_2_O are as follows: 47.46% C, 3.69% H, and 8.77% N; analysis calculations included 47.10% C, 3.93% H, and 8.68% N. The results of FTIR tests allowed the determination of the composition and structure of the synthesized complex compound [VO(dipic)(dmbipy)] · 2 H_2_O. The IR spectrum shows the highest intensity band located in the frequency range of 1600–1800 cm^−1^ and its maximum is the band for the C=O stretching vibrations. These bands are derived from esters. In the 1550–1400 cm^−1^ wavelength range, we see a high concentration of low-frequency peaks that come from the C=C double bonds located in the aromatic ring. We see many peaks of distinct intensity at the frequency of 1250–900 cm^−1^. In this area we see bands derived from the bonds between C and H, which are also found in the aromatic ring. Additionally in this range are the bands derived from the stretching vibrations between the carbon and nitrogen atoms, which participate in the formation of the heteroaromatic ring. A high-intensity band can also be seen at frequencies between 1350–1200 cm^−1^; this is a signal coming from the stretching vibrations between carbon and oxygen. This signal indicates the presence in the structure of groups derived from carboxylic acids. The IR spectrum also shows V=O vibrations bands, which occur at a frequency of about 900 cm^−1^.

The MALDI-TOF-MS method allowed the determination of the fragmentation method [VO(dipic)(dmbipy)] · 2 H_2_O into the following fragmentation ions: M^+^ (-2 H_2_O), 2 M^+^ (-dmbipy).

Both complex compounds were used as precatalysts in olefin oligomerization. The obtained products of oligomerization were subjected to FTIR, MALDI-TOF-MS, and TG analysis. In the case of 2-chloro-2-propen-1-ol oligomerization, the band with the highest absorbance in the FTIR spectra occurs at the frequency in the range 3550–3400 cm^−1^ (see Appendix A). This band confirms stretching vibrations originating from the hydroxyl group present in the chain of the oligomer. In the range of 2950–2800 cm^−1^ there is a noticeable small band indicating the presence of a C-H bond in the oligomer chain. C-C stretching vibrations are identified by the presence of a band at a frequency in the range 1600–1700 cm^−1^. Additionally, the presence of bending vibrations -CH_2_ is indicated by the presence of bands in the frequency range 1500–1450 cm^−1^. In the frequency range 1250–900 cm^−1^, bands from the C-O stretching vibrations are noticeable. On the other hand, the presence of C-Cl stretching vibrations is confirmed by the presence of bands in the range of 750–500 cm^−1^. The results of the MALDI-TOF-MS analysis performed using a matrix of α-cyano-4-hydroxycinnamic acid (CCA) and dihydroxybenzoic acid (DHB) afforded the conclusion that the oligomer obtained with the use of [VOO(dipic)](2-phepyH) · H_2_O as a precatalyst is characterized by a molecular ion with the value m/z = 179.3. This means that systems consisting of 2 mers prevail in the oligomer sample. By analyzing the spectrum, it can also be stated that in the oligomer sample there are chains with 3 mers, 5 mers, 6 mers, and 7 mers (Figure 2). This is due to the presence of peaks at the following m/z values: 235.1; 326.5; 481.1; 559.0; 712.1, respectively. The 2-chloro-2-propen-1-ol oligomer samples obtained using [VO(dipic)(dmbipy)] · 2 H_2_O as the precatalyst contain chains consisting of 5, 7, 9, 11, 13, and 15 mers.

Samples of the obtained oligomers were also examined by a thermal analysis technique coupled with FTIR (Figure 3).

This analysis showed that the thermal decomposition of the oligomer obtained using [VOO(dipic)](2-phepyH) · H_2_O as a precatalyst takes place in three steps (Figure 4). In the first step, 66.7% of the oligomer is broken down at a temperature of 198 °C. In the second stage, up to 305 °C, there is a weight loss of 7.8%. In turn, in the third step, the weight of the oligomer sample is reduced by 7.2%. As a result of thermal decomposition, 2-chloro-2-propen-1-ol oligomers release the following chemical compounds: H_2_O, CO, CO_2_, and HCl. In the case of oligomerization of 2-chloro-2-propen-1-ol carried out using [VO(dipic)(dmbipy)] · 2 H_2_O as precatalyst, the obtained oligomer sample is thermally decomposed in four steps. As a result of increasing the temperature to 120 °C, the weight of the oligomer sample decreased by 9.8%. In the next stage, the weight loss is 14.7%, which is the result of heating the sample to the temperature of 290 °C. The last two steps of thermal decomposition resulted in a weight loss of 12.9% and 7.7% as a result of heating the sample to 495 °C and 1000 °C, respectively. As for the thermal decomposition of the oligomer sample obtained with [VOO(dipic)](2-phepyH) · H_2_O as a precatalyst, the following chemicals are released: H_2_O, CO, CO_2_, and HCl (see Appendix A).

In the 2-propen-1-ol oligomerization, the IR spectrum shows that the most characteristic peak with the highest intensity and the largest half-width is that of the alcohol O-H stretching vibration, at the frequency of 3500–3400 cm^−1^ (see Appendix A). This group is involved in hydrogen bonding. At the wavenumber of around 1600 cm^−1^, there is a C-C bond signal derived from the oligomer chain. The band at the frequency of about 1450 cm^−1^ is a fairly low-intensity band coming from the -CH_2_ stretching vibrations; this group is also found in the oligomer chain. The high intensity bands in the 1230–890 cm^−1^ region are due to C-O stretching vibrations. In the MALDI-TOF-MS spectrum, the ion with the value m/z = 212.24 has the highest intensity; it is the molecular ion (Figure 2). The mass of the molecular ion shows that the investigated oligomer consists of 4 mers. The spectrum also shows peaks at 294.2, which indicates the 5 mer, 379.2 (6 mer), 461 (8 mer), 568 (9 mer), 672.0 (11 mer), 841.05 (14 mer), and 1030 (18 mer). The TG-FTIR analysis of the 2-propen-1-ol oligomer reveals that during the thermal decomposition of this oligomer, H_2_O, CO, and CO_2_ are released (see Appendix A). The bands at 4000–3500 cm^−1^ and 1500–1250 cm^−1^ indicate the O-H stretching vibrations of the water. The peaks at a frequency of 2500–2000 cm^−1^ confirm that a low absorbance band results from the C=O stretching vibrations in the CO_2_ molecule. The spectrum also shows the CO signal in the frequency of 1800–1500 cm^−1^, where C-O stretching vibrations occur.

For norbornene oligomers, it can be concluded that the most characteristic signal in the frequency of 3500–3250 cm^−1^ is a peak of very high intensity and half-width (see Appendix A). This peak is due to the O-H stretching vibration. The low-intensity peak, derived from the C-H bond in the oligomer chain, lies in the wavenumber range of 2950–2900 cm^−1^. At a frequency of approx. 1600 cm^−1^, there is a band coming from the C-C stretching vibrations. On the other hand, at a frequency of 1450 cm^−1^, we can see a band derived from -CH_2_ vibrations belonging to the oligomer chain. In the 1000–500 cm^−1^ wavelength range, peaks also originating from the C-H stretching vibrations are visible. The MALDI-TOF-MS spectrum shows that the molecular peak is 212.2 m/z, which corresponds to an oligomer of 2 mers (Figure 2). In the FTIR spectrum, we can also identify the presence of other peaks that correspond to oligomers consisting of a larger number of mers. The peak at 290.2 m/z corresponds to 3 mer, 445.1 m/z to 5 mer, 672 m/z to 7 mer, 841.0 m/z to 9 mer, and 1030.1 m/z to 11 mer. The TGA curve reveals that the thermal decomposition of the oligomer takes place in four stages. In the first disintegration step, the weight of the oligomer is reduced by 19.8% at 130 °C. In the second stage, 40% of the mass is lost at temperatures up to 245 °C, while in the third stage, the mass loss is 4% at temperatures up to 450 °C. In the last stage, the mass drops by 3.6% (see Appendix A). During the thermal decomposition of the norbornene oligomer, water, CO, and CO_2_ are released (Figure 5).

Based on literature data [35,36,37], we suggest that the mechanism of oligomerization reaction with the use of [VOO(dipic)](2-phepyH) · H_2_O and [VO(dipic)(dmbipy)] · 2 H_2_O as precatalysts is analogous to that with Ziegler–Natta catalysts. There are three stages in this type of mechanism:Stage I: formation of the catalytic system through the reaction between the complex compound (precatalyst) and the activator (MMAO-12);Stage II: matching the appropriate olefin to the active center;Stage III: incorporation of the olefin into the V-O bond by creating positive and negative partial charges.

The literature data [38,39], which are based on long-term experimental data collected on the reaction mechanism of vanadium complex compounds with a similar structure, more specifically post-metallocene complex compounds, confirm that the oligomerization reaction mechanism works in the same way as for Ziegler–Natta catalysts.

In the final part of the research, the catalytic activities (Ca) of the synthesized complex compounds as precatalysts were determined using the formula:Ca = m_o_/(n_V_ · t)
m_o_—weight of the oligomer sample [g];

n_V_—the number of mmol of vanadium(IV) or vanadium(V) ions used in the oligomerization process [mmol];

t—the time of carrying out the oligomerization process [h].

In the oligomerization of 2-chloro-2-propen-1-ol, the coordination compound [VOO(dipic)](2-phepyH) · H_2_O has a catalytic activity equal to 281.56 g · mmol^−1^· h^−1^, and the complex compound [VO(dipic)(dmbipy)] · 2 H_2_O has a catalytic activity of 522.07 g · mmol^−1^· h^−1^. In addition, in the case of 2-propen-1-ol, [VOO(dipic)](2-phepyH) · H_2_O exhibits catalytic activity equal to 171.54 g · mmol^−1^· h^−1^. However, in the norbornenen oligomerization, the dioxovanadium(V) complex has catalytic activity equal to 90.45 g · mmol^−1^· h^−1^. Using the generally accepted criterion of catalyst classification [35] due to their effectiveness, both tested complexes can be included in the group of highly active catalysts, because they have catalytic activity above 100 g · mmol^−1^· h^−1^. An exception is the value of the catalytic activity of the [VOO(dipic)](2-phepyH) · H_2_O for the oligomerization of norbornene, which is below 100 g · mmol^−1^· h^−1^.

In the literature [36] there are examples of other complex compounds of oxovanadium(IV) as 2-chloro-2-propen-1-ol precatalysts: [VO(oda)(H_2_O_2_)], [VO(oda)(bipy)] · 2 H_2_O, [VO(ida)(H_2_O)] · H_2_O, [VO(ida)(bipy)] · 2 H_2_O, and [VO(ida)(phen)] · H_2_O (Table 1).

## 4. Conclusions

The synthesized oxovandium(IV) and dioxovanadium(V) complexes in combination with the dipicolinate anion and 2-phenylpyridine and 4,4′-dimethoxy-2,2′-bipyridyl can be used as post-metallocene catalysts. In the complex compound [VOO(dipic)](2-phepyH) · H_2_O, the vanadium(V) cation is five-coordinate and has octahedral coordination geometry. On the other hand, the dipicolinate complex compound oxovanadium(IV) has a coordination number of six. The oligomerization products were analyzed using IR, MALDI-TOF-MS, TG, and TG-FTIR methods. The analyses conducted showed that the complex compounds of dioxovanadium(V) and oxovanandium(IV) after activation with MMAO-12 are active catalysts in the oligomerization process of 2-propen-1-ol, 2-chloro-2-propen-1-ol, and norbornene. The analyses also showed that the dipicolinine complex compound dioxovanadium(V) with 2-phenylpyridine has the highest catalytic activity in the oligomerization of 2-chloro-propen-1-ol. Summing up, the synthesized compounds can successfully serve as a next-generation catalyst for olefin oligomerization.

## Figures and Tables

**Figure 1 materials-15-01379-f001:**
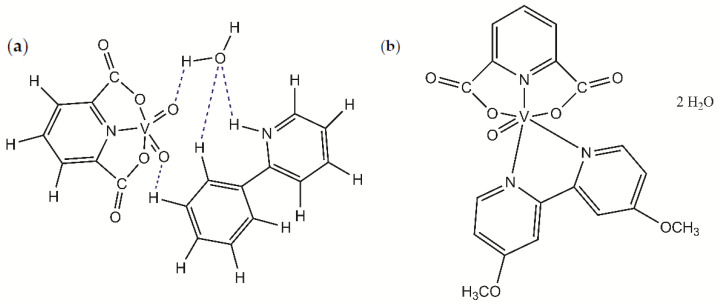
Molecular formulas of (**a**) [VOO(dipic)](2-phepyH) · H_2_O and (**b**) [VO(dipic)(dmbipy)] · 2 H_2_O.

**Figure 2 materials-15-01379-f002:**
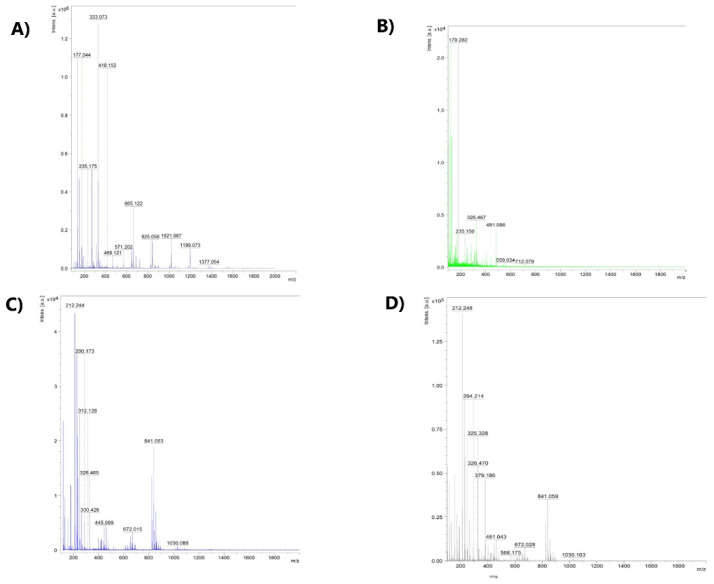
The MALDI-TOF-MS spectra of: (**A**) 2-chloro-2-propen-1-ol oligomers obtained using [VO(dipic)(dmbipy)] · 2 H_2_O, (**B**) 2-chloro-2-propen-1-ol oligomers obtained using [VOO(dipic)](2-phepyH) · H_2_O, (**C**) norbornene oligomers obtained using [VOO(dipic)](2-phepyH) · H_2_O, and (**D**) 2-propen-1-ol oligomers obtained using [VOO(dipic)](2-phepyH) · H_2_O.

**Figure 3 materials-15-01379-f003:**
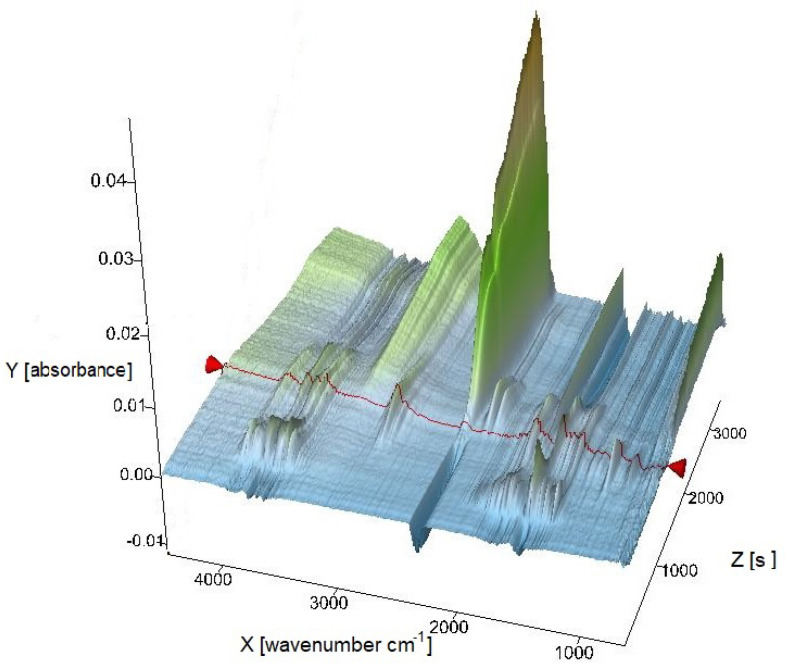
3D spectrum obtained by TG-FTIR technique for an oligomer sample synthesized using oxovanadium(IV) dipicolinate complex compound with 4,4′-dimethoxy-2,2′-bipyridyl.

**Figure 4 materials-15-01379-f004:**
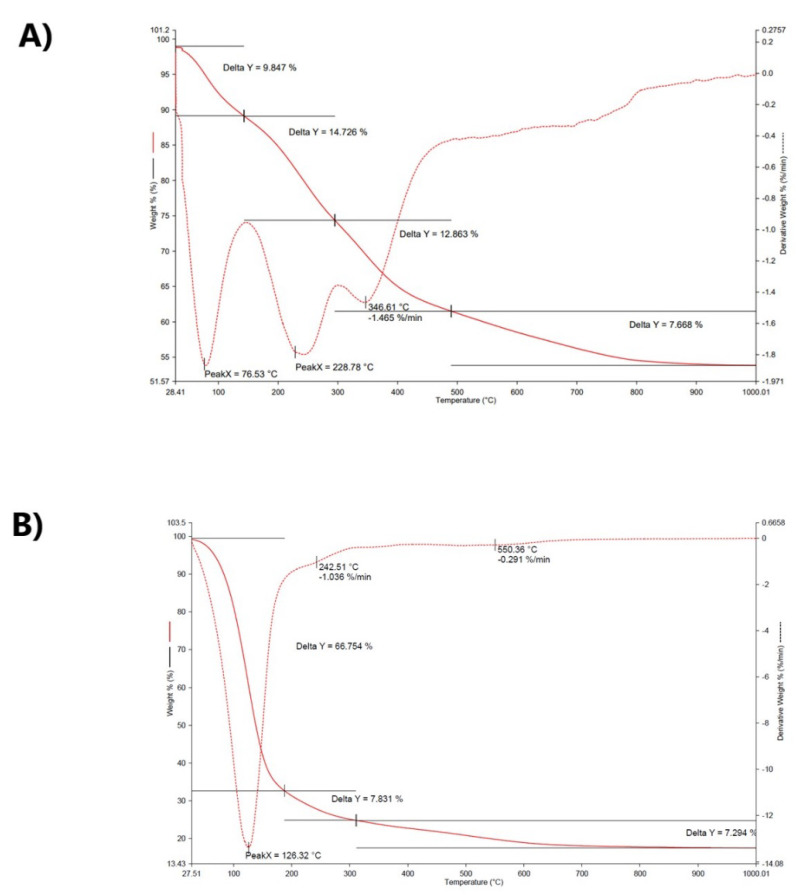
The TG of 2-chloro-2-propen-1-ol oligomers obtained using (**A**) [VO(dipic)(dmbipy)] · 2 H_2_O and (**B**) [VOO(dipic)](2-phepyH) · H_2_O.

**Figure 5 materials-15-01379-f005:**
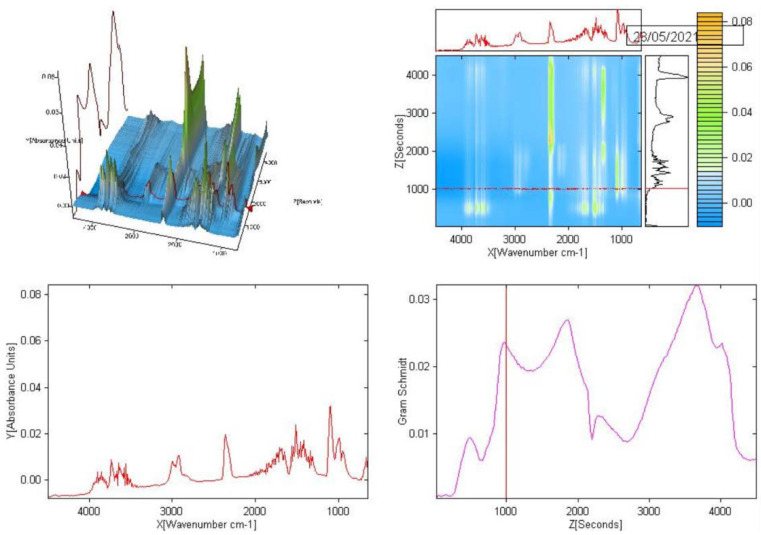
The TG-FTIR of norbornene oligomers obtained using [VOO(dipic)](2-phepyH) · H_2_O.

**Table 1 materials-15-01379-t001:** Comparison of the catalytic activity values for the compounds included in the publication and the newly synthesized compounds.

Complex Compound	Catalytic Activity [g · mmol^−1^· h^−1^]
[VOO(dipic)](2-phepyH) · H_2_O	281.56
[VO(dipic)(dmbipy)] · 2 H_2_O	522.07
[VO(oda)(H_2_O_2_)]	1004.70
[VO(oda)(bipy)]· 2H_2_O	499.53
[VO(ida)(H_2_O)]· H_2_O	799.10
[VO(ida)(bipy)]·2 H_2_O	811.13
[VO(ida)(phen)]· H_2_O	608.97

## Data Availability

Upon request of those interested.

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
