# Peer review of "Dipicolinate Complexes of Oxovanadium(IV) and Dioxovanadium(V) with 2-Phenylpyridine and 4,4′-Dimethoxy-2,2′-bipyridyl as New Precatalysts for Olefin Oligomerization"

_materials, 2022, doi:10.3390/ma15041379_

Round 1

Reviewer 1 Report

The article by Drzeżdżon et al. is devoted to the synthesis of vanadium complexes and their use as catalysts.

In its current form, this ms cannot be published in the Materials.

Metal complex catalysis remains one of the most important branches of chemistry and attracts enduring interest. However, work in this area must be done at a decent level, since adding data on one or two complexes that do not show fantastic characteristics will not tell the community much. It is desirable for research to obtain data from several new compounds and compare them with known catalysts. The work should begin with from a detailed characterization of the substances. Structural analysis of a single crystal or, at least, quantum-chemical modeling of the structure is necessary. The mechanism of the catalytic reaction should be determined or suggested. Magnetic measurements possible for vanadyl ions can tell a lot about both the reaction mechanism and the properties of the new inorganic substance. The English language needs to be seriously improved.

Reviewer questions:

Page 1 Line 26
 Compounds such as 2-propen-1-ol, 2-chloro-2-propen-1-ol and norborene are olefins.
This statement is quite obvious and not appropriate in the abstract.

P1L29 In this report, two complexes of oxovanadium(IV) and dioxovanadium(V) with dipicolinate, 2-phenylyridine and 4,4'-dimetohxy-2,2'-bipyridyl.
This sentence is missing a predicate.

P1L40 In the last 30 years, the use of organometallic compounds of the d-block elements of the periodic table, such as: vanadium, rhenium, titanium, cobalt, ruthenium, rhodium, tungsten, and cobalt has been resulted in the development of new , innovative and selective methods for the synthesis of organometallic compounds [1-4].
It would be optimal to provide references to review papers on the use of complexes of each of the listed metals in the catalysis of olefin polymerization.
P3L95 VO (acac)2
Extra space, should be [VO(acac)2]
P3L96 were dissolved in 50 ml of water
Did all these substances dissolve to form a clear solution?

Why is the crystal structure of [VO(dipic)(dmbipy)]*2H2O not determined?
The catalytic characterization appears to be satisfactory, however, publication requires more attention to graphs and tables. Buried in text data (P5L171-P8L269) is difficult to read.
P8L274 The synthesized oxowanad (IV) and dioxowanad (V)
Oxovanadium, dioxovanadium!

Author Response

Answers to the Reviewers’ comments

We are very grateful to the Reviewers for their time and constructive comments on our manuscript. We have implemented their comments and suggestions and wish to submit a revised version of the manuscript for further consideration in Materials. Changes in the initial version of the manuscript are highlighted in yellow in the revised version. Below, we also provide a point-by-point response explaining how we have addressed each of the Reviewers’ comment.

Answers to the Reviewer #1:

Comment:

               The article by Drzeżdżon et al. is devoted to the synthesis of vanadium complexes and their use as catalysts. In its current form, this ms cannot be published in the Materials. Metal complex catalysis remains one of the most important branches of chemistry and attracts enduring interest. However, work in this area must be done at a decent level, since adding data on one or two complexes that do not show fantastic characteristics will not tell the community much. It is desirable for research to obtain data from several new compounds and compare them with known catalysts. The work should begin with from a detailed characterization of the substances. Structural analysis of a single crystal or, at least, quantum-chemical modeling of the structure is necessary. The mechanism of the catalytic reaction should be determined or suggested. Magnetic measurements possible for vanadyl ions can tell a lot about both the reaction mechanism and the properties of the new inorganic substance. The English language needs to be seriously improved.

Reviewer questions:

Page 1 Line 26 Compounds such as 2-propen-1-ol, 2-chloro-2-propen-1-ol and norborene are olefins. This statement is quite obvious and not appropriate in the abstract.

Authors' Response:

            Thank you for all your suggestions. In the case of the complex compound [VO(dipic) (dmbipy)] · 2 H2O, we made numerous attempts to crystallize the substance by changing the conditions, including solvents, but we always obtained the form of a powder. For the moment, we are not able to carry out magnetic studies of vanadyl ions. However, I will definitely be looking for opportunities to perform this type of experiment. We will take it into account when planning further studies in this subject field. Based on the analysis of literature data, we have added a description of the suggested mechanism, because we assume that the precatalysts described in our report function as Ziegler-Natta catalysts:

 “Based on literature data [35-37], we suggest that the mechanism of oligomerization reaction with the use of [VOO(dipic)](2-phepyH) · H2O and [VO(dipic)(dmbipy)] · 2 H2O as precatalystsit is analogous to the Ziegler-Natta catalysts. There are three stages in this type of mechanism:

Stage I: formation of the catalytic system through the reaction between the complex compound (precatalyst) and the activator (MMAO-12);

Stage II: matching the appropriate olefin to the active center;

Stage III: incorporation of the olefin into the V-O bond by creating positive and negative partial charges.”

English has been revised and improved throughout the manuscript.

 According to the Reviewer recommendation the statement “Compounds such as 2-propen-1-ol, 2-chloro-2-propen-1-ol and norborene are olefins.” has beeen removed in the revised version of the manuscript. The fragment of the abstract has been corrected as follows:

Polyolefins are used in everyday life, they are used in the production of a large amount of plastics. In addition, polyolefins account for over 50% of the polymers produced in the world. After carrying out the oligomerization reaction of 2-propen-1-ol, 2-chloro-2-propen-1-ol and norborene, polyolefins are obtained.”

Comment:

               P1L29 In this report, two complexes of oxovanadium(IV) and dioxovanadium(V) with dipicolinate, 2-phenylyridine and 4,4'-dimetohxy-2,2'-bipyridyl. This sentence is missing a predicate.

Authors' Response:

               The sentence has been corrected:

“In this report, two complexes of oxovanadium(IV) and dioxovanadium(V) with dipicolinate, 2-phenylyridine and 4,4’-dimethoxy-2,2’-bipyridyl as precatalysts for 2-propen-1-ol, 2-chloro-2-propen-1-ol and norborene oligomerizations.”

Comment:

               P1L40 In the last 30 years, the use of organometallic compounds of the d-block elements of the periodic table, such as: vanadium, rhenium, titanium, cobalt, ruthenium, rhodium, tungsten, and cobalt has been resulted in the development of new , innovative and selective methods for the synthesis of organometallic compounds [1-4].
It would be optimal to provide references to review papers on the use of complexes of each of the listed metals in the catalysis of olefin polymerization.

Authors' Response:

               According to the Reviewer recommendation the review references were assigned to each metal confirming its use in olefin polymerization:

„In the last 30 years, the use of organometallic compounds of the d-block elements of the periodic table, such as: vanadium [1], titanium [1], rhodium [2], tungsten [3], molybdenum [3] and cobalt [1,4] has been resulted in the development of new, innovative and selective methods for the synthesis of organometallic compounds.”

Comment:

               P3L95 VO (acac)2 Extra space, should be [VO(acac)2]

Authors' Response:

               We apologize for this editing error. It has been corrected in the revised version of the manuscript.

Comment:

               P3L96 were dissolved in 50 ml of water. Did all these substances dissolve to form a clear solution?

Authors' Response:

               This part of the sentence has been corrected as follows:

“were mixed and to this mixture 50 ml of water has been added.”

Comment:

               Why is the crystal structure of [VO(dipic)(dmbipy)]·2H2O not determined?
The catalytic characterization appears to be satisfactory, however, publication requires more attention to graphs and tables. Buried in text data (P5L171-P8L269) is difficult to read.

Authors' Response:

               We made numerous attempts to crystallize the [VO(dipic)(dmbipy)] · 2 H2O, substance by changing the conditions, including solvents, but we always obtained the form of a powder.      

               In the case of  the catalytic characterization more references to specific graphs, drawings in supplementary material have been added in the revised version of the manuscript.

Comment:

               P8L274 The synthesized oxowanad (IV) and dioxowanad (V)
Oxovanadium, dioxovanadium!

Authors' Response:

               We apologize for these mistakes. It has been corrected in the revised version of the manuscript.

Reviewer 2 Report

In this manuscript, the authors described their efforts on the development of two complexes of oxovanadium (IV) and dioxovanadium (V) with dipicolinate, 2-phenylpyidine and 4.4’-dimethoxy-2,2’bipyridyl as precatalysts for olefin oligomerization. Good catalytic activity was obtained for the oligomerization of 2-chloro-2-propen-1-ol and 2-propen-1-ol. The result is interesting. However, in comparison to their previous work (ref 36), the current two complexes did not exhibit obvious advantages over other oxovanadium complexes. It seems to be an extension of their previous work. Moreover, there are some typos in the manuscript need to be carefully checked. In my opinion, the interest of the current research lies in the development of novel catalysts, which might be applied to other reactions, despite the unsatisfactory results in this work. From the view of this point, it might be suitable for the publication in Materials.

Author Response

Answers to the Reviewer #2:

Comment:

               In this manuscript, the authors described their efforts on the development of two complexes of oxovanadium (IV) and dioxovanadium (V) with dipicolinate, 2-phenylpyidine and 4.4’-dimethoxy-2,2’bipyridyl as precatalysts for olefin oligomerization. Good catalytic activity was obtained for the oligomerization of 2-chloro-2-propen-1-ol and 2-propen-1-ol. The result is interesting. However, in comparison to their previous work (ref 36), the current two complexes did not exhibit obvious advantages over other oxovanadium complexes. It seems to be an extension of their previous work. Moreover, there are some typos in the manuscript need to be carefully checked. In my opinion, the interest of the current research lies in the development of novel catalysts, which might be applied to other reactions, despite the unsatisfactory results in this work. From the view of this point, it might be suitable for the publication in Materials.

Authors' Response:

            Thank you very much for these valuable comments. We will take it into account when planning further studies in this subject field. In the introduction part in the revised version of the manuscript we have added the information that this report is a continuation of our previous work on olefin oligomerization catalysis. We sincerely apologize for the linguistic errors. All typos have been corrected. Changes in the initial version of the manuscript are highlighted in yellow in the revised version.

Reviewer 3 Report

Dipicolinate complexes of oxovanadium(IV) and dioxovanadium(V) with 2-phenylpyridine and 4,4'-dimethoxy-2,2'-bipyridyl as new precatalysts  for olefin oligomerization

From the similarity check using Turnitin, the similarity level of this article is quite high, reaching 24%. This indicates that the originality of the article is questionable. In terms of topics, this article is actually quite good, the scientific discussion is quite comprehensive. The task of the author now is to lower the similarity level to below 15%. The authors should discuss the novelty and state of the art of this research in depth in the introduction section.

From the analysis point of view, it should be noted that the instrumentations used to reveal the product structure are only 3 (elemental analysis, FT-IR and The MALDI-TOF-MS , but the amount of data generated are quite a lot (from the supplementary). In order for the discussion to be more interesting and reliable, the author should add some more analytical data in the discussion section of the article. Basically, more analytical data are needed to be displayed in articles that can support the hypothesis and conclusion.

The relationship between title, abstract, introduction, results, discussions and conclusions (IMRAD) is quite well supported by relevant references and related to the title of the article.

In terms of language used, grammar, phrases and choice of the words are quite good and easy to understand.

Author Response

Answers to the Reviewer #3:

Comment:

               From the similarity check using Turnitin, the similarity level of this article is quite high, reaching 24%. This indicates that the originality of the article is questionable. In terms of topics, this article is actually quite good, the scientific discussion is quite comprehensive. The task of the author now is to lower the similarity level to below 15%. The authors should discuss the novelty and state of the art of this research in depth in the introduction section.

Authors' Response:

            Thank you very much for these valuable comments. After analyzing the anti-plagiarism report, we can conclude that most of the phrases marked as similar to other materials available on the Internet result from the commonly accepted chemical nomenclature, such as “2-chloro-2-propen-1-ol”, “2-propen-1-ol”, “2- chloro-2-propen-1-ol oligomers”, or phrases frequently used to describe various physicochemical properties of materials such as "As a result of thermal decomposition", "peak with the highest intensity", “results of elemental analysis…”.  According to common practice, such elements should not be classified as "plagiarism". According to the Reviewer recommendation the novelty of our report has been added in the introduction section:

“The novelty of this report is the presentation of a new, so far undescribed dipicolinate complex compound of oxovanadium(IV) with 4,4'-dimethoxy-2,2'-bipyridyl, and in this manuscript for the first time we describe the use of two complex compounds: [VOO(dipic )](2-phepyH) · H2O and [VO(dipic)(dmbipy)] · 2 H2O for the oligomerization of 2-chloro-2-propen-1-ol, 2-propen-1-ol and norbornene. To the best of our knowledge, there is no information in the literature about the catalytic properties of the complexes: [VOO(dipic )](2-phepyH) · H2O and [VO(dipic)(dmbipy)] · 2 H2O.”

Comment:            

               From the analysis point of view, it should be noted that the instrumentations used to reveal the product structure are only 3 (elemental analysis, FT-IR and The MALDI-TOF-MS , but the amount of data generated are quite a lot (from the supplementary). In order for the discussion to be more interesting and reliable, the author should add some more analytical data in the discussion section of the article. Basically, more analytical data are needed to be displayed in articles that can support the hypothesis and conclusion.

Authors' Response:

            We agree with the reviewer's opinion that the more analytical data obtained with various instrumental methods, the more reliable the results of the research are. However, in the case of the studies described by us in this report, the analysis of the product by 3 methods is, in our opinion, sufficient because the purpose of our research was not the structural studies of oligomerization products, but the determination of the catalytic activity of two complex compounds: [VOO(dipic)](2-phepyH) · H2O and [VO(dipic)(dmbipy)] · 2 H2O. To better visualize the characteristics of the products more references to specific graphs, drawings in Supplementary Material have been added in the revised version of the manuscript. We characterized the obtained products with methods that were available and feasible for us. When planning future research, we will have in mind to extend the research on product characteristics with even more experimental methods.

Comment:

               The relationship between title, abstract, introduction, results, discussions and conclusions (IMRAD) is quite well supported by relevant references and related to the title of the article.

Authors' Response:

            Thank you very much.

Comment:

               In terms of language used, grammar, phrases and choice of the words are quite good and easy to understand.

Authors' Response:

            Thank you very much.

Round 2

Reviewer 1 Report

The authors have somewhat improved the manuscript compared to the first edition. The bibliography has been expanded, some clarifications have been made to the Synthesis and Results&Discussion.

However, in my opinion, the article still does not look good enough to be published in the Materials.

Perhaps my assessment is too harsh, since the other reviewer turned out to be supportive. However, in my opinion, some improvements are critically needed for publication:

1) If the complex does not form single crystals, then authors can try to solve the structure from powder diffraction data. If this is not possible, then there are methods of quantum chemical modeling could be a solution

2) Figure 1 (or rather scheme 1) must be replaced. If the binding of water and phenylpyridine by hydrogen bonding is supposed, then it should be showed in the scheme.

3) Figure 2 is the only one (sic!) experimental result presented in the main text of the article in graphical form. This significantly complicates the perception of the text.

4) Figures with mass spectrometry data from the appendix cannot be easily understood without a long careful reading of the text and constant switching between the article and the SI.

5) The appearance of a text fragment about the catalytic mechanism is very appropriate. However, what are the experimental grounds for believing that it is the same as in the literature? I admit that this may be obvious to narrow specialists in the field, however, Materials is a journal for a wide range of readers.

6) English language and style still need improvement.

Author Response

Answers to the Reviewer’s comments

We are very grateful to the Reviewer for constructive comments on our manuscript. We have implemented comments and suggestions and wish to submit a revised version of the manuscript for further consideration in Materials. Changes in the initial version of the manuscript are highlighted in yellow in the revised version. Below, we also provide a point-by-point response explaining how we have addressed each of the Reviewers’ comment.

Answers to the Reviewer #1:

Comment:

               The authors have somewhat improved the manuscript compared to the first edition. The bibliography has been expanded, some clarifications have been made to the Synthesis and Results&Discussion. However, in my opinion, the article still does not look good enough to be published in the Materials. Perhaps my assessment is too harsh, since the other reviewer turned out to be supportive. However, in my opinion, some improvements are critically needed for publication:

1) If the complex does not form single crystals, then authors can try to solve the structure from powder diffraction data. If this is not possible, then there are methods of quantum chemical modeling could be a solution

Authors' Response:

            Thank you for these valuable suggestions and we agree with the Reviewer's opinion. However, at the present stage, we do not have the possibility to perform powder XRD and quantum-mechanical theoretical research. We would have to look for cooperation in this area, but we will certainly do so having regard to our future research. However, taking into account current version of the results presented in the manuscript, the elemental analysis studies (very high compliance with the theoretical data based on the assumed composition of the complex) confirm the composition of the synthesized complex, and additionally FTIR tests confirm the presence of functional groups present in the complex and the presence of vanadium(IV) cation bonds and other elements. In addition, the results of the MALDI-TOF-MS study repeated the molecular ion.

Comment:

               2) Figure 1 (or rather scheme 1) must be replaced. If the binding of water and phenylpyridine by hydrogen bonding is supposed, then it should be showed in the scheme.

Authors' Response:

               According to the Reviewer recommendation the Figure 1 has been replaced:

Figure 1. Molecular formulas of 1) [VOO(dipic)](2-phepyH) · H2O and 2) [VO(dipic)(dmbipy)] · 2 H2O

Comment:

               3) Figure 2 is the only one (sic!) experimental result presented in the main text of the article in graphical form. This significantly complicates the perception of the text.

Authors' Response:

               According to the Reviewer recommendation three additional figures have been added to facilitate the reception of the text:

Figure 2. The MALDI-TOF-MS spectra of: A) 2-chloro-2-propen-1-ol oligomers obtained using [VO(dipic)(dmbipy)] · 2 H2O, B) 2-chloro-2-propen-1-ol oligomers obtained using [VOO(dipic)](2-phepyH) · H2O, C) norbornene oligomers obtained using  [VOO(dipic)](2-phepyH) · H2O, and D) 2-propen-1-ol oligomers obtained using  [VOO(dipic)](2-phepyH) · H2O.

Figure 4. The TG of 2-chloro-2-propen-1-ol oligomers obtained using A) [VO(dipic)(dmbipy)]                       · 2 H2O and B) [VOO(dipic)](2-phepyH) · H2O.

Figure 5. The TG-FTIR  of norbornene oligomers obtained using  [VOO(dipic)](2-phepyH) · H2O.

Comment:

               4) Figures with mass spectrometry data from the appendix cannot be easily understood without a long careful reading of the text and constant switching between the article and the SI.

Authors' Response:

               To make the text easier to read and understand the mass spectrometry data of the oligomers have been collected and presented in the revised version of the manuscript as Figure 2.

Figure 2. The MALDI-TOF-MS spectra of: A) 2-chloro-2-propen-1-ol oligomers obtained using [VO(dipic)(dmbipy)] · 2 H2O, B) 2-chloro-2-propen-1-ol oligomers obtained using [VOO(dipic)](2-phepyH) · H2O, C) norbornene oligomers obtained using  [VOO(dipic)](2-phepyH) · H2O, and D) 2-propen-1-ol oligomers obtained using  [VOO(dipic)](2-phepyH) · H2O.

Comment:

               5) The appearance of a text fragment about the catalytic mechanism is very appropriate. However, what are the experimental grounds for believing that it is the same as in the literature? I admit that this may be obvious to narrow specialists in the field, however, Materials is a journal for a wide range of readers.

Authors' Response:

               The literature data [38, 39], which are based on the long-term experimental data collected on the reaction mechanism of vanadium complex compounds with a similar structure, more specifically post-metallocene complex compounds, confirm that the oligomerization reaction mechanism works in the same way as for Ziegler-Natta catalysts. We added this information in the revised version of the manuscript.

Comment:

               6) English language and style still need improvement.

Authors' Response:

               English has been improved.         

Round 3

Reviewer 1 Report

The article has been improved and may be published.